**Data Availability Statement:** CHAPAO is freely available as an open source software at https://github.com/ashiq24/CHAPAO. All the datasets

# CHAPAO: Likelihood and hierarchical reference-based representation of biomolecular sequences and applications to compressing multiple sequence alignments

Md Ashiqur Rahman[ID]◉, Abdullah Aman Tutul◉, Sifat Muhammad Abdullah◉, Md. Shamsuzzoha Bayzid[ID]*

Department of Computer Science and Engineering/Bangladesh University of Engineering and Technology, Dhaka, Bangladesh

◉ These authors contributed equally to this work.
* shams_bayzid@cse.buet.ac.bd

## Abstract

### Background

High-throughput experimental technologies are generating tremendous amounts of genomic data, offering valuable resources to answer important questions and extract biological insights. Storing this sheer amount of genomic data has become a major concern in bioinformatics. General purpose compression techniques (e.g. gzip, bzip2, 7-zip) are being widely used due to their pervasiveness and relatively good speed. However, they are not customized for genomic data and may fail to leverage special characteristics and redundancy of the biomolecular sequences.

### Results

We present a new lossless compression method CHAPAO (**CO**mpressing **A**lignments using **H**ierarchical and **P**robabilistic **A**pproach), which is especially designed for multiple sequence alignments (MSAs) of biomolecular data and offers very good compression gain. We have introduced a novel hierarchical referencing technique to represent biomolecular sequences which combines likelihood based analyses of the sequence similarities and graph theoretic algorithms. We performed an extensive evaluation study using a collection of real biological data from the avian phylogenomics project, 1000 plants project (1KP), and 16S and 23S rRNA datasets. We report the performance of CHAPAO in comparison with general purpose compression techniques as well as with MFCompress and Nucleotide Archival Format (NAF)—two of the best known methods especially designed for FASTA files. Experimental results suggest that CHAPAO offers significant improvements in compression gain over most other alternative methods. CHAPAO is freely available as an open source software at https://github.com/ashiq24/CHAPAO.

analyzed in this paper are from previously published studies and are publicly available. Avian datasets were collected from the GigaScience repository, GigaDB (http://gigadb.org/dataset/101041). The 16S and 23S datasets are available at https://sites.google.com/eng.ucsd.edu/datasets/alignment/16s23s?authuser=0. The 1KP dataset was obtained from https://datacommons.cyverse.org/browse/iplant/home/shared/onekp_pilot.

**Funding:** The author(s) received no specific funding for this work.

**Competing interests:** The authors have declared that no competing interests exist.

## Conclusion

CHAPAO advances the state-of-the-art in compression algorithms and represents a potential alternative to the general purpose compression techniques as well as to the existing specialized compression techniques for biomolecular sequences.

## Background

One of the major tasks of bioinformatics is to collect, analyze and interpret large volumes of biomolecular data. The amount of available genomic data is increasing approximately tenfold every year, at a much faster rate than Moore's Law for computational power [1, 2]. This advancement in sequencing technologies demands more efficient ways to store and analyze large genomic datasets. Numerous general purpose compression algorithms, such as zip and gzip based on DEFLATE algorithm [3], bzip2 using Burrows-Wheeler transform [4], 7-zip [5] are being widely used to deal with the genomic data deluge. However, these general purpose compression techniques are agnostic about the special characteristics and redundancy existing in the biomolecular sequences. Thus, due to the growing awareness about the challenges posed by the genomic data deluge and the inability of the general purpose compression techniques to take advantage of the redundancy in genomic data, developing specialized compression techniques for biomolecular sequences has drawn substantial attention from the bioinformatics community.

Biomolecular sequence compression has been an active research area over the last decade. Most of these works are focused on directly compressing individual DNA/RNA sequences, such as BioCompress [6, 7], GenCompress [8], the CTW+LZ algorithm [9], DNACompress [10], MFCompress [11], DELIMINATE [12], XM [13], Pinho *et al.* [14] and Tabus and Korodi [15]. This class of methods utilizes various properties of genomic sequences such as small alphabet size and repetitive regions. There is another class of compression techniques, known as reference-based methods, that takes advantage of the redundancy in the biomolecular sequences. Here, a reference sequence is used to encode a "target sequence", resulting in substantial compression when there are significant similarities between the reference sequence and the target sequence. Reference-based approach is a popular technique for genomic data compression, and has been used in many methods including RLZ [16], GRS [17], GReEn [18], coil [6], Fritz *et al.* [19], Christley *et al.* [20], Brandon *et al.* [21], Wang and Zhang [17], Kozanitis *et al.* [22] and Popitsch *et al.* [23]. These methods are useful in compressing sequence databases or storing millions of reads produced by next generation sequencing technologies.

Multiple sequence alignment (MSA) is the alignment of biological sequences, inferring homologies by reflecting basic evolutionary events (insertion, deletion, and substitution). Constructing an MSA is a basic step in many analyses in computational biology such as phylogenetic tree construction, orthology identification, predicting the structure, and function of proteins. Therefore, an exponentially increasing number of MSA files are being generated and analyzed in various domains of computational biology. This underscores the need for developing methods for the efficient storage of MSA files. However, there has not been notable advancement in developing compression techniques that are especially customized to consider the special characteristics and redundancy of MSAs.

Fundamental to the recent advancements in compressing sequence data is the ability to leverage the redundancy of the biomolecular sequences [6, 19]. Likewise, MSA files have specific formats and characteristics. Hickey *et al.* [24] proposed a way for saving MSA files on the

basis of phylogenetic hierarchy. Matos *et al*. presented a model using a special arithmetic coding for DNA multiple sequence alignment blocks [25]. Many of these existing studies aimed more at presenting a concept than at providing publicly available usable compression tools. Moreover, many of them are dependent on external reference sequences [18–20, 22] which limits their practical use. Furthermore, some of them can handle only the four-letter alphabet (A, T, C, G), preventing their applications to protein sequences. Thus, although the last two decades have witnessed the proposal of many algorithms for compressing genomic sequences, this community is still dependent on the general purpose compressors.

In this study, we present CHAPAO, a reference-based technique for compressing MSA files. This is to our knowledge the first application of the reference-based technique for compressing MSAs. Unlike conventional reference-based methods where an "extra" sequence (not included in the input sequence to compress) is used as a reference [17–20, 22], we proposed a novel *hierarchical referencing* technique where a suitable subset of the input sequences in the MSA file is used as reference sequences. Our referencing technique is hierarchical in a sense that a subset $S_1$ of sequences can be used to encode a subset $S_2$ of sequences, and $S_2$ can subsequently be used to encode another subset of sequences. Thus, we aim to keep an optimal subset of input sequences that can encode all the sequences in the MSA in a hierarchical manner. We have proposed a likelihood based technique to model the sequence similarity and "representability", and subsequently apply a minimum spanning arborescence [26–28] based algorithm on a graph modeled from the MSA in order to find an optimal set of reference sequences and an optimal order of hierarchical referencing.

We performed an extensive evaluation study to assess the performance of CHAPAO on the MSA files from the Avian Phylogenomics [29, 30] and 1000 plants (1KP) [31, 32] projects (two of the largest phylogenomics projects to date) containing various types of gene sequences (introns, exons, and UCEs). We also analyzed a collection of large and challenging ribosomal RNA datasets (16S and 23S) obtained from the Gutell Lab [33, 34]. In addition to the general purpose compressors (zip, gzip, bzip2, and LZMA [35] (implemented in the 7-zip archiver [36])), we compared with MFCompress [11], which is the best known alternative method for compressing FASTA files, and Nucleotide Archival Format (NAF) [37]. Experimental results suggest that CHAPAO offers notable compression gain and significantly outperforms the best alternate methods except for 7-zip.

## Methods

### Overview of CHAPAO

In conventional reference-based techniques, the target sequences (sequences to be compressed) are represented in terms of a reference sequence and some additional metadata information. The additional information may be insertion, substitution, or deletion from reference sequence, that will convert the reference sequence to the target sequence. Fig 1 shows two sequences that evolved with substitutions, insertions, and deletions and the corresponding multiple sequence alignment. We denote a pair of reference *r* and target *t* sequences by a tuple <*r*, *t*>. Usually, only one reference sequence is used for all the target sequences. This works well when the sequences come from the same or closely related species (as it is the case for Fritz *et al*. [19] where they used a reference genome sequence to map the short reads). For compressing an MSA with sequences from a collection of species with higher degrees of dissimilarity between them, single reference based techniques may result in higher amounts of metadata, and may lead to lower compression ratio.

CHAPAO finds a suitable subset of sequences in the MSA as reference. Unlike other reference-based techniques [18–20, 22], CHAPAO does not depend on any external reference

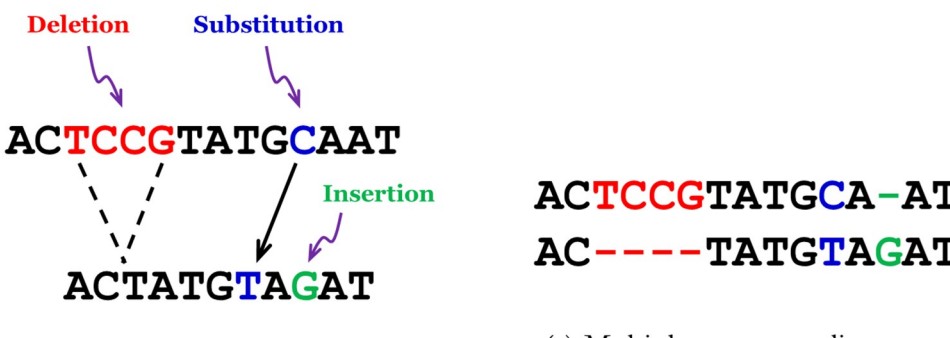

**Fig 1. Character evolution and multiple sequence alignment.** (a) Two observed sequences, (b) Character evolution with substitution and indels which can change the sequence length and blur the homology, and (c) Multiple sequence alignment of the two sequences capturing the underlying character evolution where each site consists of homologous characters.

sequence. An efficient statistical and graph theoretic algorithm has been incorporated in CHA-PAO to find an optimal set $R$ of reference sequences so that other (target) sequences $T$ can be *hierarchically* obtained from $R$ with minimum *representational cost*. This is a hierarchical approach where a subset $T_1 \subseteq T$ is encoded using $R$, and subsequently $T_i$ ($i > 1$) is encoded using sequences from $R \cup T_1 \cup \ldots \cup T_{i-1}$.

In order to find an optimal set of reference sequences, we create an *encodability graph* $\mathcal{EG}$ where each vertex corresponds to a sequence and the weight $w_{i,j}$ of a directed edge $(S_i, S_j)$ from $S_i$ to $S_j$ represents the *cost* of representing sequence $S_j$ by sequence $S_i$. Next, we find a *minimum spanning arborescence* $\mathcal{MA}$ in $\mathcal{EG}$ using Edmond's algorithm [38, 39]. This $\mathcal{MA}$ defines an optimal set of reference sequences and hierarchical referencing ($<r, t>$) relationships among the sequences (see Theorem 0.1). Appropriate metadata are generated to decode the sequence *hierarchically* from the reference sequences. Note that the encodability graph $\mathcal{EG}$ would be a very dense graph—a directed complete graph where every pair of vertices is connected by a pair of edges (one in each direction). To keep $\mathcal{EG}$ relatively sparse, edges are established only between the nodes that correspond to "similar" sequences. We used a likelihood based approach to find similar sequences. Fig 2 shows an overview of the algorithmic workflow of CHAPAO. We used bzip2 at the final stage of our algorithm to compress the reference sequences and the metadata.

## Representational cost

The cost $C_{i,j}$ of representing a target sequence $S_j$ using a reference sequence $S_i$ depends on the metadata required to store in order to retrieve $S_j$ from $S_i$. $C_{i,j}$ includes the cost of storing the indices of the positions where $S_i$ and $S_j$ differ, and the cost of storing the mismatched bases. Thus, $C_{i,j} = f(N, I)$, where, $N$ is the number of bits required to store a base, and $I$ is the number of bits required to store an index. Fig 3(b) shows the cost matrix $\mathcal{M}_c$, showing the cost of representing every pair of sequences (in both directions) in Fig 3(a). Note that $C_{2,1} = 2I + 4N$, but

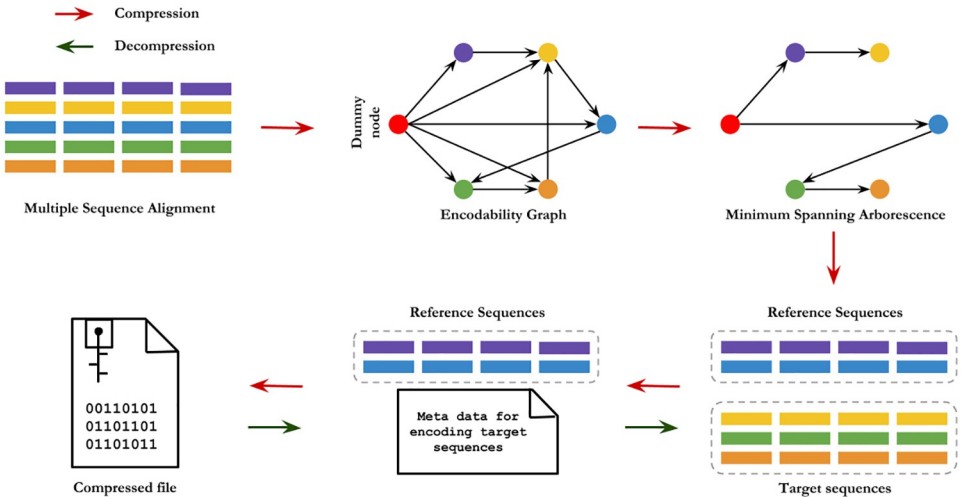

**Fig 2. Overview of the compression and decompression techniques in CHAPAO.** A directed weighted "Encodability Graph" is constructed where each vertex corresponds to a sequence in MSA except for the dummy node (shown in red) which is used as a "source" vertex. Next, a minimum spanning arborescence $\mathcal{MA}$ in the graph is constructed. Sequences that are children of the dummy node in the $\mathcal{MA}$ will be used as reference sequences. Appropriate metadata are generated to hierarchically represent all other sequences. Finally, the reference sequences along with the metadata are compressed using existing compression techniques. The pipeline is completely reversible, allowing lossless decompression of the original MSAs.

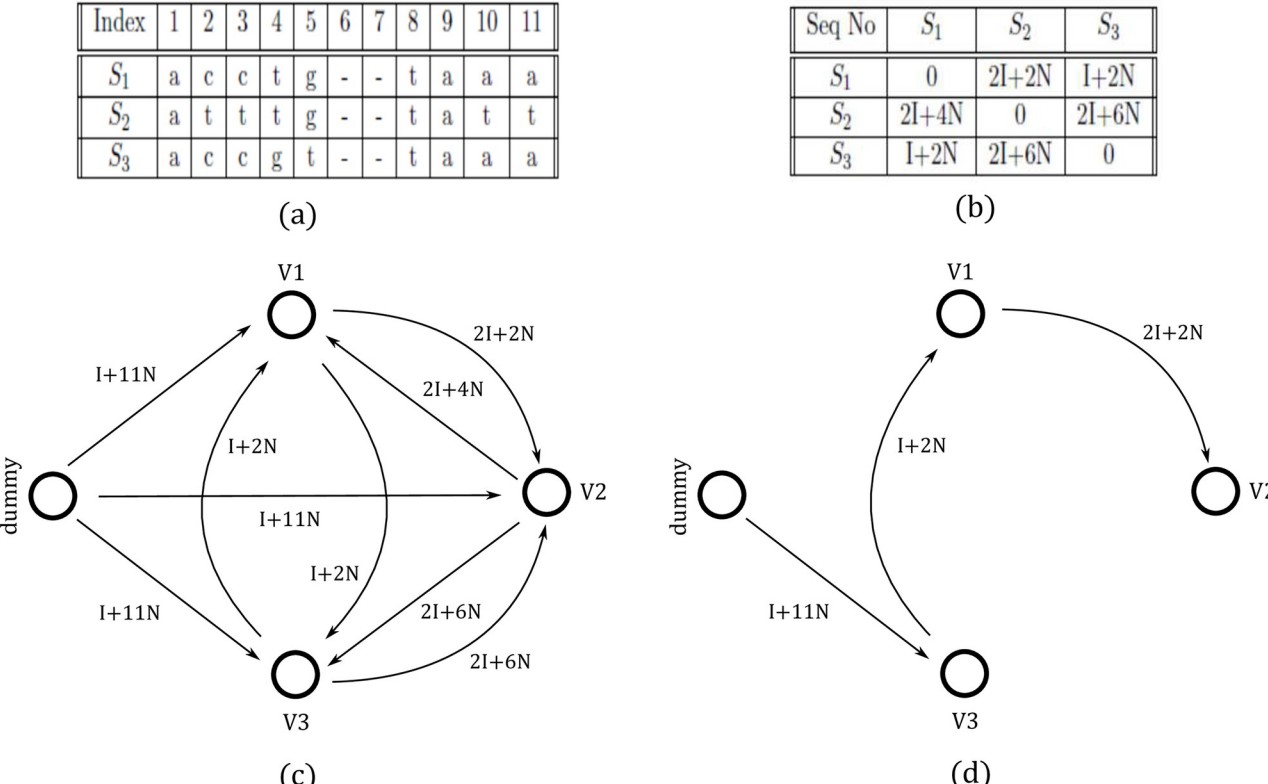

**Fig 3. Directed graph based modeling.** (a) A multiple sequence alignment with three sequences, (b) the corresponding cost matrix, (c) the encodability graph $\mathcal{EG}$, and (d) the corresponding minimum spanning arborescence $\mathcal{MA}$.

$C_{1,2} = 2I + 2N$. These two sequences differ in indices 2, 3, 10, and 11. To represent $S_2$ using $S_1$, we need to store "tt" and indices 2 and 10, whereas we need to store "aa" at 2 and "cc" at 10 for representing $S_1$ using $S_2$. Therefore, $C_{i,j}$ is not necessarily identical to $C_{j,i}$ and thus the encodability graph is a directed complete graph. The cost of storing a reference sequence is $I + l^*N$, where $l$ is the length of the sequence, and $I$ is the cost to store the index of the sequence. We store the index of a sequence to ensure that the sequences in the decompressed MSA are in exactly the same order as in the original uncompressed MSA.

## Modeling the encodability graph

Given an MSA with $n$ sequences, we create an encodability graph $\mathcal{EG}$ with $n + 1$ vertices where $n$ vertices correspond to the $n$ sequences in the MSA. The $(n + 1)$-th vertex is a dummy vertex $v_d$ which does not correspond to any sequence in the MSA. There is an edge from $v_d$ to $i$ ($1 \le i \le n$), where $w_{d,i}$ represents the cost of storing sequence $S_i$ as a reference sequence. Thus, the dummy node acts as the "source" vertex in the $\mathcal{EG}$, and a minimum spanning arborescence $\mathcal{MA}$ is constructed considering the dummy node as the root node. Thus, in addition to the representational cost $w_{i,j}$, the cost for storing a reference sequence is considered in the encodability graph, and therefore, the minimum spanning arborescence $\mathcal{MA}$ in $\mathcal{EG}$ defines the optimal set of reference sequences and an optimal order of hierarchical referencing (see Theorem 0.1). A sequence $S_i$ is considered to be a reference if there is an edge $(v_d, S_i)$ from $v_d$ to $S_i$ in the $\mathcal{MA}$. The hierarchical referencing is defined by the directed edges in $\mathcal{MA}$. Fig 3(c) shows the encodability graph of the MSA shown in Fig 3(a), and the corresponding $\mathcal{MA}$ is shown in Fig 3(d). Here, the cost of storing all three DNA sequences would be the summation of the cost to save sequence $S_3$ as a reference, the cost to represent $S_1$ using $S_3$ as a reference, and the cost to represent $S_2$ using $S_1$ as a reference. Thus, we only need to store $S_3$ and appropriate metadata to hierarchically decode $S_1$ and $S_2$.

White and Hendy [6] previously used an undirected-graph based technique to model the similarity among the sequences in a database. They used edit tree distance as an approximation of the maximum parsimony distance to find groups of similar sequences. They split the whole database into multiple undirected similarity graphs composed of highly similar sequences. Next, (undirected) minimum spanning trees are computed for each of the similarity graphs. The smallest sequence in each similarity graph is considered as the only reference sequence for all the target sequences. Therefore, despite some parallels, our technique—with the directed encodability graph based approach using likelihood based similarity and subsequent computation of minimum spanning arborescence and the hierarchical referencing—is significantly different than the one used in White and Hendy [6].

**Theorem 0.1**. *The minimum spanning arborescence $\mathcal{MA}$ in an encodability graph $\mathcal{EG}$ defines an optimal set of reference sequences and reference-target ($< r, t >$) relationships.*

*Proof.* Let $\mathcal{R}$ be the set of reference sequences and $\mathcal{RT}$ be the set of reference-target pairs suggested by $\mathcal{MA}$. Let $C$ be the total weight of $\mathcal{MA}$, meaning that $C$ is the cost of storing $\mathcal{R}$ and $\mathcal{RT}$. Assume that $C$ is not optimal, meaning that there exists another set $\mathcal{R}'$ of reference sequences and a set $\mathcal{RT}'$ of reference-target pairs that can be stored with cost $C'$, and $C' < C$. Let us build a graph $\mathcal{M}^{A'}$ where a vertex corresponds to a sequence in $\mathcal{R}'$ and there is an edge from $S_i$ to $S_j$ if $< S_i, S_j > \in \mathcal{RT}'$. The weight of an edge $(v_i, v_j)$ represents the cost of storing $S_j$ using $S_i$ as a reference. Finally, we add a dummy node $v_d$ to $\mathcal{M}^{A'}$ and add edges from $v_d$ to all the reference sequences in $\mathcal{R}'$, where the cost of an edge represents the cost of storing a reference sequence. It is easy to see that $\mathcal{M}^{A'}$ is a spanning arborescence, rooted at $v_d$, of $\mathcal{EG}$ with

cost $C'$. Therefore, since $C' < C$, $\mathcal{MA}$ cannot be a minimum spanning arborescence, which leads to a contradiction. This completes the proof.

## Log-likelihood based similarity modeling

For a directed complete graph, calculating the cost matrix $\mathcal{M}_c$ is expensive and requires $O(n^2 l)$ time where $n$ is the number of sequences and $l$ is the length of each sequence. For computational efficiency, CHAPAO tries to keep the encodability graph relatively sparse by considering only those edges that are incident on reasonably similar sequences. However, finding pairwise similar sequences is computationally expensive as well. Therefore, we have introduced an efficient heuristic using the likelihood values of the sequences. Let $\mathcal{M}$ be a multiple sequence alignment with $n$ sequences. A sequence $S_i$ in $\mathcal{M}$ can be considered as an $l$-dimensional random vector, $S_i = [S_{i1}, S_{i2}, \ldots, S_{il}]$, where $S_{ij} \in \{a, t, g, c, -\}$ refers to the $j$-th base (character) in $S_i$. Thus, $\mathcal{M}$ is a collection of $n$ $l$-dimensional random vectors. We assume that the occurrence of a base at a position $j$ in a sequence $S_i$ is independent of any other base in $S_i$. Therefore, the likelihood of the sequence $S_i$ in $\mathcal{M}$ can be computed as follows.

$$\mathcal{L}(\mathcal{M}|S_i) = p(S_i|\mathcal{M}) = \prod_{j=1}^{j=l} p(S_{ij}|\mathcal{M}). \tag{1}$$

Here, $p(S_{ij}|\mathcal{M})$ is the probability of the occurrence of a particular base $S_{ij} \in \{a, t, c, g, -\}$ at column $j$ in $\mathcal{M}$. Let $\mathcal{F}_{S_{ij}}$ be the number of times $S_{ij}$ appears at column $j$ in $\mathcal{M}$. Then $p(S_{ij}|\mathcal{M})$ can be calculated as follows.

$$p(S_{ij}|\mathcal{M}) = \frac{\mathcal{F}_{S_{ij}}}{n}. \tag{2}$$

As the individual probability values are very small, we take the log-likelihood as follows.

$$\log p(S_i|\mathcal{M}) = \sum_{j=1}^{j=l} \log p(S_{ij}|\mathcal{M}). \tag{3}$$

We sort the sequences in an MSA according to their likelihood values so that the adjacent sequences in the sorted list have a relatively low cost for representing each other. Next we take a sliding window of a preferred length $l_w$ (which is a tunable parameter), and slide it over the sorted list. The step size (sliding length) is chosen appropriately to ensure a certain amount of overlap between two windows. The sequences within a window will form a clique (i.e. every pair of vertices will be connected with each other) in the encodability graph $\mathcal{EG}$. This reduces the time complexity of computing the cost matrix $\mathcal{M}_c$ to $O(nl)$.

## Time complexity

The time complexity of our algorithm depends on the cost of computing the cost matrix and computing the minimum spanning arborescence $\mathcal{MA}$. For each edge $e$ in the encodability graph $\mathcal{EG}$, we have to calculate its weight which takes $O(l)$ time. Thus, the time complexity for computing the cost matrix will be $O(El)$, where $E$ is the number of edges in $\mathcal{EG}$.

For an MSA with $n$ sequences, a sliding window of length $l_w$ and step size (sliding amount) $l_s$, there will be $\lceil \frac{n-l_w}{l_s} \rceil + 1$ number of cliques in $\mathcal{EG}$. Time complexity to compute the cost matrix for a clique is $l_w^2 l$ as there are $l_w^2$ edges in a clique with $l_w$ nodes. Considering all the cliques in $\mathcal{EG}$, the time complexity is $\sum_{i=1}^{\lceil \frac{n-l_w}{l_s} \rceil + 1} l_w^2 l$. Note that $\left( \frac{n-l_w}{l_s} + 1 \right) \approx O\left( \frac{n}{l_s} \right)$, and $l_w = cl_s$,

where $c$ is a positive real number. Thus, the time complexity of computing the cost matrix is as follows.

$$\sum_{i=1}^{\lceil \frac{n}{l_s} \rceil} l_w^2 l = \lceil \frac{n}{l_s} \rceil l_w^2 l \approx n l_s l c^2.$$

As $l_s$ is a constant which is usually much smaller than the length of the sequences ($l$), and does not depend on $n$, computing the cost matrix takes $O(nl)$ time. This also implies that the number of edges in the encodability graph, constructed by considering the overlapping windows of sequences with similar likelihood values, is $O(n)$. We implemented Edmond's algorithm [38, 39] to find the minimum spanning arborescence, which takes $O(VE)$ time. Thus, the time complexity of our compression pipeline is $O(El + VE)$. Therefore, without the log-likelihood based heuristic version with sliding windows, CHAPAO takes $O(n^2 l + n^3)$ time. But the compression time is reduced to $O(nl + n^2)$ using our proposed sparse graph representation, saving a factor of $O(n)$.

## Experimental studies

We evaluated the performance of CHAPAO on a collection of real and challenging biological datasets. We used data from two of the largest phylogenomics projects to date: 1) Avian phylogenomics project [29, 30] and 2) 1000 plants (1KP) project [31, 32]. We also analyzed two other widely used large biological datasets (16S and 23S) from Gutell Lab [33, 34, 40] containing alignment files with large numbers of sequences from 16S and 23S ribosomal RNA sampled from bacteria. S1 Table in S1 File shows the summary of various alignments in these datasets. We assessed the performance of CHAPAO on DNA sequence alignments.

We compared CHAPAO with several popular general purpose compression techniques, namely zip, bzip2, gzip, and LZMA [35] (implemented in the 7-zip archiver [36]). We also compared CHAPAO with special purpose compression techniques, MFCompress [11] and NAF [37], which is especially targeted to compress biomolecular sequences in FASTA files. MFCompress was previously compared with gzip, bzip2, ppmd (a variant of ppm [41]), and LZMA as well as with the recent special purpose compressor DELIMINATE [12], and was shown to be the best method for compressing FASTA files. NAF is based on zstd [42] and was shown to achieve a compression ratio close to DELIMINATE.

In order to compare various compression techniques, we report the average compression gains (over all the MSAs in a particular dataset) attained by different methods. We also show the size of the MSAs after compression by different methods, and we divide the MSA files in a particular dataset into different bins based on the size of the MSAs to better assess the performance of different method across varying file sizes. We performed Wilcoxon signed-rank test (with $\alpha = 0.05$) to measure the statistical significance of the differences between two methods. The experiments were performed on a Windows machine with an Intel Core I7–7500U processor (3.5 GHz), 8GB DDR4 RAM, and 128GB SSD memory (SATA 3).

The performance of CHAPAO may vary depending on the window size. Longer window sizes are expected to achieve better compression gain at the cost of more compression time. We used window sizes ranging from $20 - 40$ on various datasets. These smaller window sizes provided enough compression gain to significantly outperform most other methods. The particular window size and overlap size that we used to generate the results are mentioned in each subsequent figure.

## Results on avian dataset

Avian phylogenomics project is the largest vertebrate phylogenomics project [29, 30], which assembled or collected the genomes of 48 avian species spanning most orders of birds. This dataset contains exons from 8251 syntenic protein-coding genes, introns from 2516 of these genes, and a nonoverlapping set of 3769 ultraconserved elements (UCEs). The exon gene set was prepared based on synteny-defined orthologs chosen from the assembled genomes of chicken and zebra finch. The intron gene set consists of 2516 genes that are orthologous subset of introns from the 8295 protein-coding genes. The UCE dataset has 3679 genes with ∼1000 bp of flanking sequences. The UCE dataset was filtered to remove overlap with the exon and intron datasets.

Fig 4 shows the relative performance of various methods on avian dataset. Since we have thousands of alignments covering a wide range of file sizes, we show the results for various bins of different file size limits. CHAPAO consistently achieved a significantly higher compression ratio than all other methods, except LZMA and NAF, regardless of the file size and sequence type. LZMA, despite being a general purpose compression technique, achieved the best compression gains on all the model conditions on avian datasets, followed by CHAPAO and NAF.

CHAPAO and NAF achieved competitive compression gain on Intron datasets, while NAF was slightly better than CHAPAO on the UCEs and CHAPAO was slightly better than NAF of the Exons. On the Intron alignments, CHAPAO achieved 38.8%, 31.3%, 14.3% and 16.6% more compression than zip, gzip, bzip2, and MFCompress respectively (see Fig 4(b)). CHAPAO achieved the second best compression ratio on exon MSAs, where it achieved 20.28% more compression than MFCompress, 6.2% more compression than NAF, and 24.64% more compression than bzip2 (see Fig 4(c)). On the UCE dataset, CHAPAO achieved 21.6% more compression than MFCompress and 15.9% more than bzip2, and NAF achieved slightly better compression (2.29%) than CHAPAO (see Fig 4(a)).

To assess the applicability and performance of our method on very large alignments, we analyzed the concatenated alignments resulting from concatenating the alignments of introns, exons and UCEs. We do not analyze the ultra-large alignments as the likelihood based analysis is computationally intensive for very large alignments, and restricted our analyses to the files not exceeding 300 MB [43] (see S2 and S3 Tables in S1 File). Although concatenation (also known as combined analysis) can be problematic as it is agnostic to the topological differences among the gene trees [44–49], it is one of the most widely used methods for species tree estimation from multi-locus data. Therefore, there is intrinsic value in storing data of this nature. Similar to individual gene sequence alignments, LZMA achieved the best compression gain on concatenated alignments as well except for MSA-7 and MSA-8, where CHAPAO achieved the best compression gains (Fig 5). However, the performance of NAF substantially degraded on these large concatenated alignments. CHAPAO and MFCompress achieved second best compression gains. On an average, CHAPAO achieved 45.15% better compression than bzip2 which is the second best performing general purpose compressor on the concatenated alignments. Unlike other datasets, MFCompress outperformed CHAPAO on six (out of 10) MSAs. However, among the largest three MSAs (MSA-7, -8, and -9), CHAPAO outperformed MFCompress on two of them (MSA-7 and MSA-8). On the other large file (MSA-9), MFCompress is better than CHAPAO. We investigated the average *p*-distance of the sequences in these large MSA files and observed that the average *p*-distance of MSA-9 is 0.15 which is much higher than those of MSA-7 and MSA-8 (0.043 and 0.029 respectively), indicating a lower level of similarity/redundancy in MSA-9 compared to MSA-7 and MSA-8. This could explain why CHAPAO did not achieve the same level of compression gain on MSA-9 as it did on MSA-7

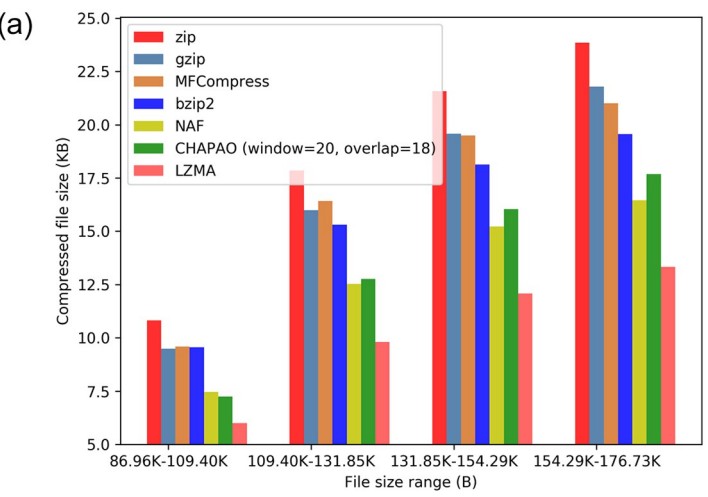

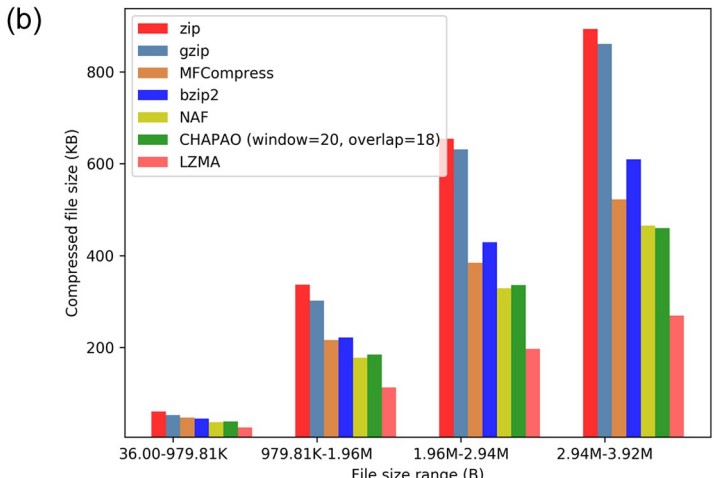

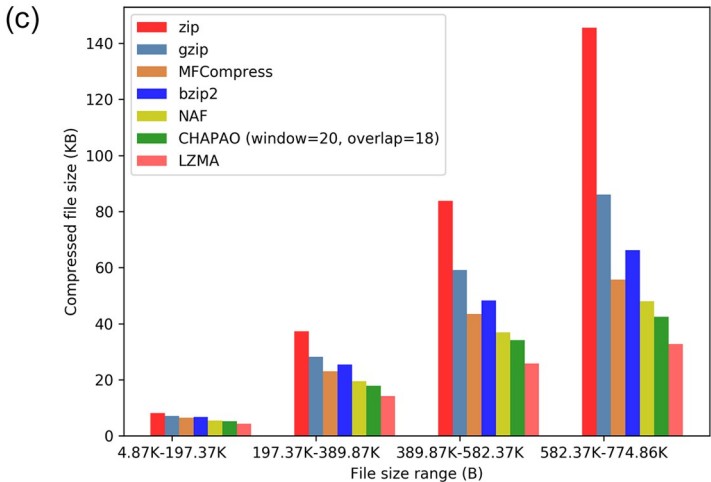

**Fig 4. Performance of various compression techniques on avian datasets.** To better understand the relative performance of different methods across different file sizes, we distribute the MSA files into various bins based on their sizes. For each bin (file-size range), we show the average size of the compressed files produced by various methods. (a) UCEs. (b) Introns. (c) Exons.

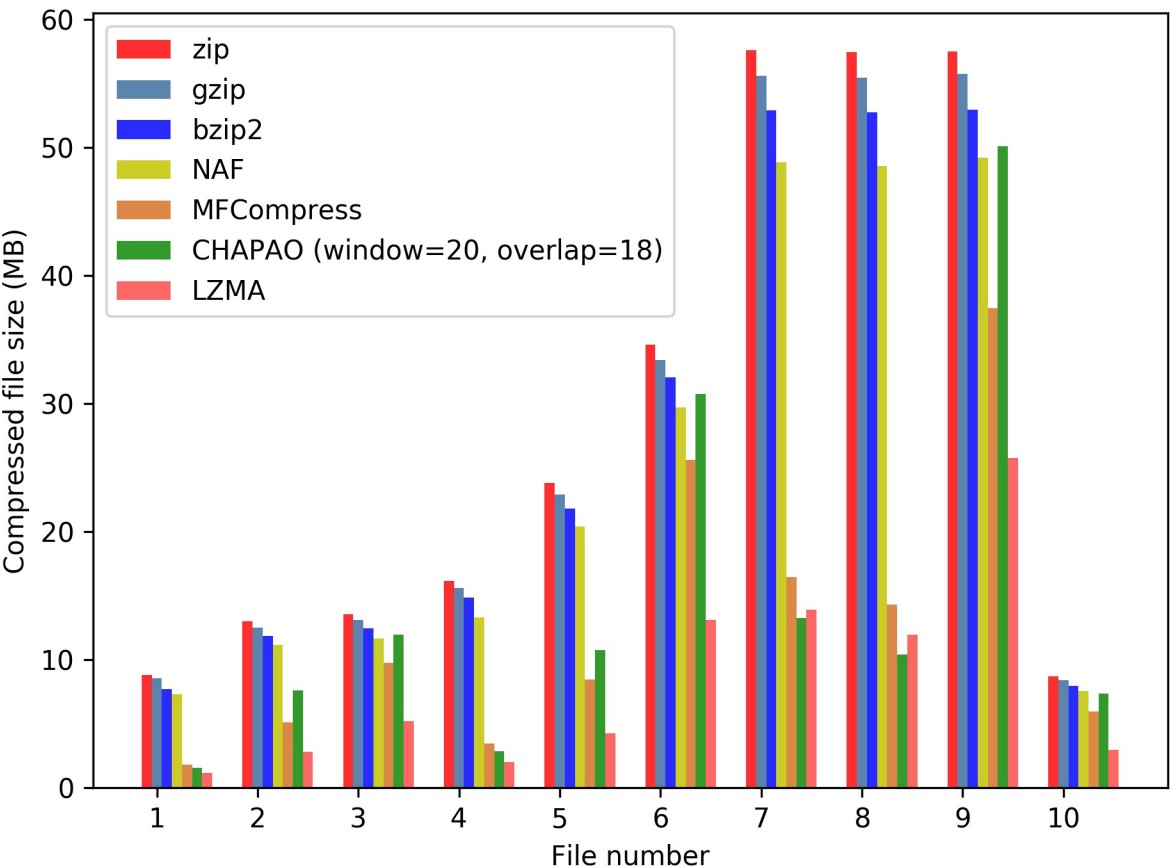

**Fig 5. Performance of various compression techniques on 10 concatenated alignments in avian dataset.**

and MSA-8. Note also that both CHAPAO and MFCompress achieved substantially better compression gain than bzip2 on MSA-7 and MSA-8, but the gain is not that substantial on MSA-9. These results suggest that CHAPAO can effectively capture the similarities/redundancies in the sequences, and hence underscore the importance of using special purpose compressor for large scale MSAs in order to effectively leverage the redundancies in biomolecular sequences. CHAPAO significantly outperformed NAF on seven (out of 10) MSAs. The compression gains of CHAPAO and NAF are competitive on the remaining three files. Notably, on an average, CHAPAO achieved 40.81% more compression than NAF on these concatenated alignments.

The avian dataset is distributed as gzip-compressed files which consumes 923 MB (considering only the MSAs analyzed in this study) [43]. However, CHAPAO can archive these files using 604 MB of data, saving 34.56% of the storage requirement. Moreover, this compression gain has been achieved using a window size of 20 and can be further improved by using longer window sizes at the cost of more computation time.

There is a notable correlation between the similarity/redundancy in MSA files and the compression ratio obtained by CHAPAO, which is in line with our objective of leveraging the similarity in the biomolecular sequences. We investigated this on avian datasets. We computed the hamming distances between the pairs of sequences in every MSA file in the avian datasets. We have defined the average hamming distance of an MSA file according to Eq 4. Here, $N$ and $L$ are the number of sequences and the length of each sequence in an MSA file, respectively.

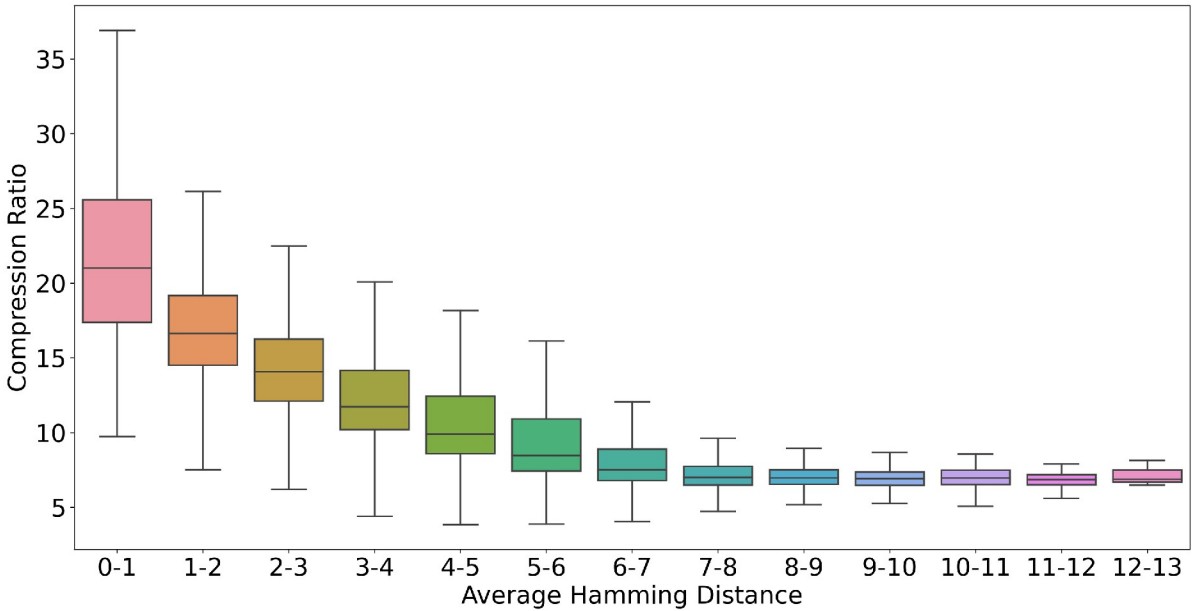

**Fig 6. Performance of CHAPAO with varying levels of dissimilarity/divergence of the 14,490 MSAs in avian datasets.** The average hamming distance (as defined in Eq 4) of these files ranges from 0–13. The box plots show the compression ratio (ratio of the size of the original file and the compressed file) of CHAPAO on the MSAs in avian datasets (here MSAs are sorted in an ascending order of their average hamming distance).

*Hamming*($S_i$, $S_j$) denotes the hamming distance between two sequence $S_i$ and $S_j$. CHAPAO has achieved more compression on the MSAs with less average pairwise hamming distance among the sequences (Fig 6). As the level of dissimilarity between the pairs of sequences in an MSA file is increased, the compression gain of CHAPAO gradually decreases. The experimental results also suggest that CHAPAO may provide better compression gain for MSAs with relatively large numbers of sequences. For MSAs with large numbers of sequences, relatively larger proportions of the sequences may be expressed as non-reference sequences, which subsequently leads to higher compression gain.

$$\text{Average Hamming Distance} = \frac{(\sum_{i=1}^{N-1} \sum_{j=i+1}^{N} Hamming(S_i, S_j))/L}{N} \quad (4)$$

One of the notable observations from these results is the superior performance of general purpose compression technique LZMA compared to various special purpose compression techniques (e.g., MFCompress, CHAPAO, NAF). The LZ algorithms approach the data sequentially and keep track of all the strings that appeared in the past up to a certain limit (window size) [50, 51]. If the current substring is previously seen, it is then replaced by a reference to the previous occurrence. LZMA is an improvement of LZ coding which can detect repeats that are further apart. As a result, it can capture both intra-sequence and inter-sequence similarities [52], whereas CHAPAO tries to leverage only the inter-sequence similarities. We believe that this could be a reason why the performance of CHAPAO is not better than LZMA in most cases.

## Results on 16S and 23S datasets

Results on 16S and 23S datasets are shown in Fig 7a and 7b. LZMA and CHAPAO achieved extra-ordinary compression gain on these datasets. MFCompress performed worse than bzip2

even though it is a special purpose tool for compressing FASTA files. LZMA was the best performing method on 23S dataset, and CHAPAO was the second best method, which achieved 80%, 78.68%, 55.38%, 65.47%, and 45.84% more compression than zip, gzip, bzip2, MFCompress, and NAF respectively. Similar trends hold for 16S dataset, where CHAPAO obtained 76.76%, 73.26%, 32.82%, 68.25%, and 33.92% more compression than zip, gzip, bzip2, MFCompress and NAF respectively. CHAPAO was able to compress the 16S and 23S datasets, originally occupying 361.66 MB and 58.79 MB respectively, to only 7.2 MB and 1.3 MB which can easily be transmitted as an email attachment.

## Results on 1KP dataset

The 1000 plants (1KP) initiative has generated large-scale gene sequencing data for over 1000 species of plants, representing approximately one billion years of evolution, including flowering plants, conifers, ferns, mosses, and streptophyte green algae [31, 32]. This dataset comprises 9,609 multiple sequence alignments each containing sequences from 1000 different plant species with a wide range of sequence lengths (303 ∼ 63150 bp). The performance of CHAPAO along with other compression methods is shown in Fig 7(c). LZMA and CHAPAO achieved the best compression gains on this dataset, and CHAPAO obtained 42.7%, 40.3%, 7.3%, 24.1%, and 6.26% more compression than zip, gzip, bzip2, MFCompress, and NAF, respectively. Notably, CHAPAO was significantly better than LZMA on larger MSAs (243MB—284MB range), and competitive with LZMA on other size ranges.

## Impact of sliding window lengths on compression gain

Sliding window length $l_w$ and sliding amount $l_s$ are important hyper-parameters of our algorithm. With a sliding window of lengths of $l_w$, the encodability graph will be composed of a series of cliques each of size $l_w$. Smaller sizes of the sliding window may result in relatively lower compression gain. Maximum compression is expected to be obtained when the window length is equal to the number of sequences in an MSA. We investigated the impact of varying lengths of window and overlap. Fig 8 shows the impact of various hyper-parameter settings on 16S and 23S datasets. These results suggest that as we increase the window size, the compression gain tends to improve. A sliding window of length 30 with overlap of 20 (CHAPAO (window = 30, overlap = 20)) achieved almost 17.71% and 16.1% more compression than CHAPAO with a sliding window of size 5 with overlap length 3 (CHAPAO (window = 5, overlap = 3)) on 16S and 23S datasets, respectively. This impact is even more prominent on the MSAs with larger numbers of sequences. For example, CHAPAO (window = 30, overlap = 20) is 20.6% better than CHAPAO (window = 5, overlap = 3) on 16S.B.ALL which contains 27,643 sequences, whereas the improvement is 10.13% on 16S.M which contains 901 sequences. However, this improved compression gain comes with an additional computational burden. CHAPAO(window = 30, overlap = 20) took almost 3.8 times more compression time than CHAPAO(window = 5, overlap = 3) on 16S and 23S datasets. The average running times for different lengths of window and overlap on 16S and 23S datasets are shown in Table 1 (see also S11 Table in S1 File).

## Compression and decompression time

Compression times of CHAPAO (for the window sizes used in this study) and other methods are shown in S4-S12 Tables in S1 File. CHAPAO tends to take more time for compression, especially on the larger files, than other methods. However, shorter window sizes can substantially reduce the compression time. However, the decompression step is much faster and takes less computational time than MFCompress (see S15 Table in S1 File). Even for the largest ones

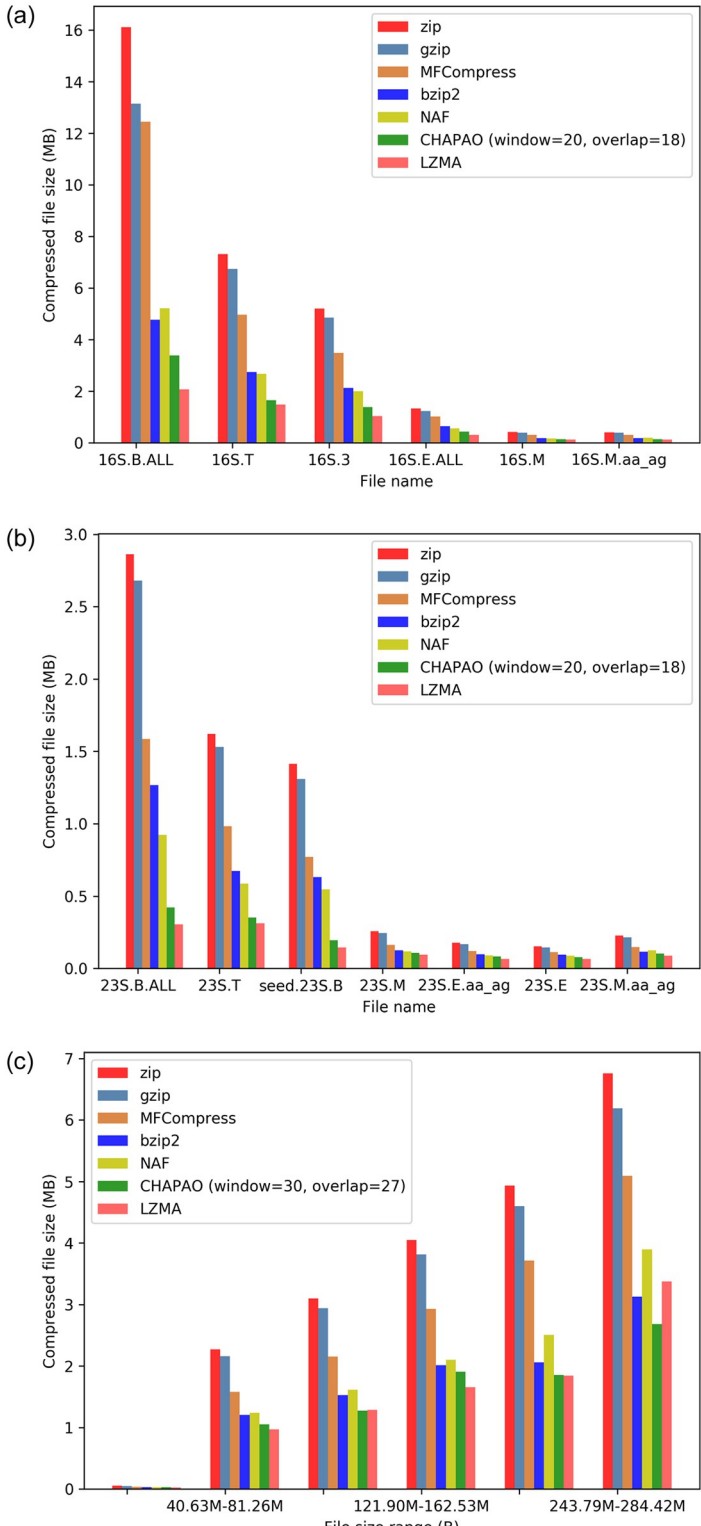

**Fig 7. Comparison of various compression techniques on 16S and 23S datasets and 1KP dataset.** (a) 16S. (b) 23S. (c) 1KP.

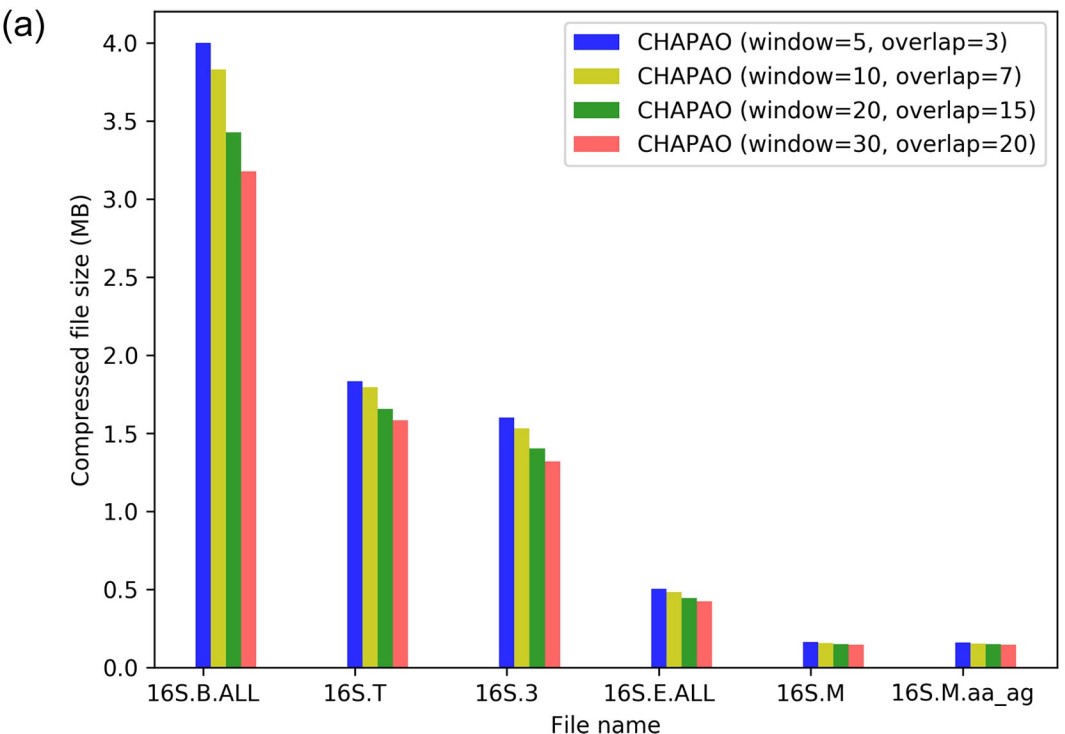

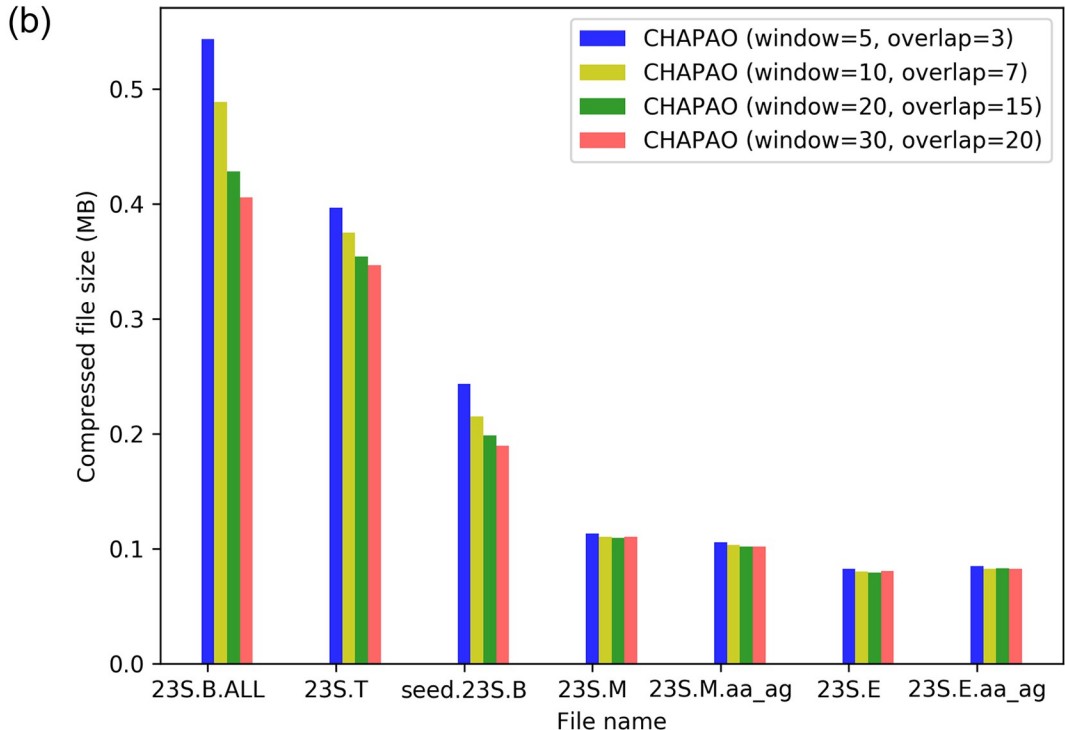

**Fig 8. Impact of the lengths of sliding window and overlap on compression ratio.** We show the performance of various variants of CHAPAO on 16S and 23S datasets. (a) 16S. (b) 23S.

**Table 1. Impact of sliding window and overlap lengths on compression time.** We show the running time of various variants of CHAPAO on 16S and 23S datasets.

| Window | Overlap | Average Running Time |
|---|---|---|
| 5 | 3 | 83.91 |
| 10 | 7 | 136.05 |
| 20 | 15 | 238.82 |
| 30 | 20 | 323 |

like 16.S.B.All, it takes only around 9 seconds, whereas MFCompress takes around 24 seconds to compress and 26 seconds to decompress (S15 Table in S1 File). NAF and LZMA (7-zip) are faster than CHAPAO both in terms of compression and decompression speed.

## Conclusions

Given the huge number of multiple sequence alignments that can be harvested from various comparative genomics projects, there is a need for efficient tools to archive them. General purpose techniques are being widely used to archive MSA files. However, these methods are agnostic to the specificity of MSAs. In this paper, we have attempted to advance the state-of-the-art in MSA compression by taking the redundancy and specificity of MSAs into account. We have presented CHAPAO, a new lossless compression technique which is especially tailored to leverage the redundancy of genomic sequences by exploiting a novel hierarchical reference-based sequence representation to allow parsimonious storage of MSAs. Extensive experimental studies on a variety of real biological datasets suggest that CHAPAO can achieve substantially higher compression gain over the existing general purpose compression techniques (except 7-zip) as well as the special purpose techniques for genomic data at the cost of more compression time. However, this study can be extended in several directions. Although CHAPAO can handle reasonably large MSAs, this is not yet scalable to ultra-large whole genome alignments due to computational requirements (time and memory). Designing appropriate divide-and-conquer frameworks—which will operate on smaller blocks in the ultra large alignments—to boost the performance of CHAPAO both in terms of scalability and compression gain would be interesting. CHAPAO cannot take in account the intra-sequence similarity like LZMA (7-zip). Capturing the similarity within a particular sequence may improve the performance of CHAPAO and so future studies need to investigate this. CHAPAO, in its current form, can run on protein sequence alignments, but CHAPAO is not particularly tailored for protein alphabets. As the number of different characters present in protein sequences is significantly higher than the alphabet size of DNA sequences, special customization is required to handle alignments of protein families (as was done in CoMSA [53]). Besides the proposed likelihood-based technique, exploring other techniques for capturing the similarity and redundancy in protein MSAs are required to identify appropriate approaches for protein MSAs. Although we have performed an extensive simulation study on various datasets with a wide range of model conditions, future studies need to analyze more datasets to further investigate the relative strengths and weaknesses of various methods to guide the users in choosing the right compressors for different datasets. Finally, our proposed hierarchical referencing technique is expected to be of potential interest for efficiently compressing short reads generated by the next generation sequencing technologies, which we leave as future work. Finally, CHAPAO represents a notable contribution towards designing compression algorithmic frameworks for biomolecular sequences and should be considered as a potential alternative to the widely used general purpose compression techniques.

## Supporting information

**S1 File. Supplementary results and data.**
(PDF)

## Author Contributions

**Conceptualization:** Md Ashiqur Rahman, Abdullah Aman Tutul, Sifat Muhammad Abdullah, Md. Shamsuzzoha Bayzid.

**Methodology:** Md Ashiqur Rahman, Abdullah Aman Tutul, Sifat Muhammad Abdullah, Md. Shamsuzzoha Bayzid.

**Software:** Md Ashiqur Rahman, Abdullah Aman Tutul, Sifat Muhammad Abdullah.

**Supervision:** Md. Shamsuzzoha Bayzid.

**Validation:** Md Ashiqur Rahman, Abdullah Aman Tutul, Sifat Muhammad Abdullah, Md. Shamsuzzoha Bayzid.

**Visualization:** Md Ashiqur Rahman, Abdullah Aman Tutul, Sifat Muhammad Abdullah, Md. Shamsuzzoha Bayzid.

**Writing – original draft:** Md Ashiqur Rahman, Abdullah Aman Tutul, Sifat Muhammad Abdullah, Md. Shamsuzzoha Bayzid.

**Writing – review & editing:** Md Ashiqur Rahman, Abdullah Aman Tutul, Sifat Muhammad Abdullah, Md. Shamsuzzoha Bayzid.

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
