## [Decision Letter · Decision Letter 0]

9 Nov 2021

PONE-D-21-27907CHAPAO: likelihood and hierarchical reference based representation of biomolecular sequences and applications to compressing multiple sequence alignmentsPLOS ONE

Dear Dr. Bayzid 

Thank you for submitting your manuscript to PLOS ONE. After careful consideration, we feel that it has merit but does not fully meet PLOS ONE’s publication criteria as it currently stands. Therefore, we invite you to submit a revised version of the manuscript that addresses the points raised during the review process.

We look forward to receiving your revised manuscript.

Kind regards,

Ram Kumar Sharma, Ph.D

Academic Editor

PLOS ONE

Journal Requirements:

Additional Editor Comments:

Both the reviewers indicated the concerns on the current version of the manuscript.

Reviewers' comments:

Reviewer's Responses to Questions

**Comments to the Author**

1. Is the manuscript technically sound, and do the data support the conclusions?

Reviewer #1: Yes

Reviewer #2: Yes

2. Has the statistical analysis been performed appropriately and rigorously? 

Reviewer #1: Yes

Reviewer #2: Yes

3. Have the authors made all data underlying the findings in their manuscript fully available?

Reviewer #1: Yes

Reviewer #2: Yes

4. Is the manuscript presented in an intelligible fashion and written in standard English?

Reviewer #1: Yes

Reviewer #2: No

5. Review Comments to the Author

Reviewer #1: The manuscript entitled "CHAPAO: likelihood and hierarchical reference based representation of biomolecular sequences and applications to compressing multiple sequence alignment", by Rahman et al, seems to be very interesting, wherein authors have put good efforts in developing a tool for compressing multiple sequence alignment. The manuscript is nicely written incorporating proper statistical tools in support of CHAPAO. I would like to recommend the manuscript, however, some typos should be corrected in the manuscript, as an instance 'arborescenc' in keywords.

Reviewer #2: Comments to the Author

This is an interesting article presenting a new compression method CHAPAO (Compressing Alignments using Hierarchical and Probabilistic Approach) especially designed for compressing MSAs. I think it contributes towards further alternatives to easily compress and decompress multiple sequence alignments (MSAs) of biomolecular data using hierarchical referencing technique combined likelihood-based analysis. Also, the algorithm was evaluated using various real biological datasets, and can be used as alternative to other compressing techniques. Moreover, several issues described below need to be addressed to improve the quality of this manuscript.

Major comments:

1. In section 3 Experimental studies: What was the optimum window size used to get better compression gain and also less compression time as the authors used 20-40 window size on various datasets

2. In section 3.1 please explain more about the datasets used such as UCS ultra-conserved elements locus sets.

3. In result section intron, exon, UCE alignment in various real datasets could you please explain the how the percentage gain achieved in CHAPAO, on comparing with four other programs. Have you considered each file bin size for deducing percentage?

4. Could you please define how this method will perform if the sequences are highly divergent, and consisting of insertion and deletion which are larger than 50bp.

5. Is this method also considering the protein sequences alignments also?

6. Edit the references according to journal format.

7. Please add author contributions section.

Minor comments:

English language: The grammar throughout the manuscript needs editing and the language in general needs some streamlining.

Introduction:

1. Page 2, Paragraph-1, Line-2, Reference based should be reference-based.

2. Paragraph-3, Line-4, etc should be etc,

3. Parahraph-4, Line-10, format should be formats.

4. Paragraph-4, Line-17, last two decades should be the last two decades.

5. Paragraph-6, Line-7, dataset should be datasets.

Methods:

1. Section 2.1, Line-1, reference based should be reference-based.

2. Line 3, substitution should be substitution,

Experimental studies:

1. Paragraph-3, Line-3. Ram should be RAM,

Section 3.1 Results on avian dataset

Paragraph 1, Line-3, set should be set,

Section 3.4 Impact of sliding window lengths on compression gain.

Paragraph-1, Line-7, tend should be tends.

Paragraph-1, Line-14, additional should be an additional.

Conclusion:

Line 7, reference based should be reference-based.

Line-21, DNA sequence should be DNA sequences.

All supplementary data and figures should be quoted in same format.

6. PLOS authors have the option to publish the peer review history of their article (what does this mean?). If published, this will include your full peer review and any attached files.

Reviewer #1: No

Reviewer #2: No

---

## [Author Response · Author response to Decision Letter 0]

28 Jan 2022

December 23, 2021

Dr. Emily Chenette

Editor-in-Chief

PLoS ONE

Dear Dr. Chenette,

Thank you and the Academic Editor, Dr. Ram Kumar Sharma for handling our manuscript and for the constructive reviews by the reviewers. We have revised the manuscript by addressing the reviewers’ comments. The new/modified material in the manuscript is provided in blue text to make it easy to identify. Detailed responses to the individual reviews are provided inline. 

The reviewers' questions led us to perform more experiments and provide additional details, and – we think – the manuscript is much improved now. We very much thank you and the reviewers for pointing out these issues. We hope the revised version will satisfy the reviewers and also meet PLoS ONE’s requirements.

Yours sincerely,

Dr. Md. Shamsuzzoha Bayzid, Corresponding Author

Department of Computer Science and Engineering

Bangladesh University of Engineering and Technology

Email: shams_bayzid@cse.buet.ac.bd

Reviewer #1: 

The manuscript entitled "CHAPAO: likelihood and hierarchical reference based representation of biomolecular sequences and applications to compressing multiple sequence alignment", by Rahman et al, seems to be very interesting, wherein authors have put good efforts in developing a tool for compressing multiple sequence alignment. The manuscript is nicely written incorporating proper statistical tools in support of CHAPAO. I would like to recommend the manuscript, however, some typos should be corrected in the manuscript, as an instance 'arborescenc' in keywords.

Response: 

Thank you very much for appreciating our effort and considering this manuscript publishable. We are very sorry for the typos. We made a sincere effort to fix this type of typos.

Reviewer #2: Comments to the Author

This is an interesting article presenting a new compression method CHAPAO (Compressing Alignments using Hierarchical and Probabilistic Approach) especially designed for compressing MSAs. I think it contributes towards further alternatives to easily compress and decompress multiple sequence alignments (MSAs) of biomolecular data using hierarchical referencing technique combined likelihood-based analysis. Also, the algorithm was evaluated using various real biological datasets, and can be used as alternative to other compressing techniques. Moreover, several issues described below need to be addressed to improve the quality of this manuscript.

Response: 

Thank you very much for your encouraging comments and for considering CHAPAO a useful tool for compressing MSAs. We appreciate your nice suggestions, which we have taken into account (please see our responses below). I hope you will find the revised manuscript publishable.

Major comments:

1. In section 3 Experimental studies: What was the optimum window size used to get better compression gain and also less compression time as the authors used 20-40 window size on various datasets

Response: 

We have already indicated, in each figure, the particular window size that we used to generate the reported results. We have now added the following sentence in Section 3 to make it clearer.

“The particular window size and overlap size that we used to generate the results are indicated in each subsequent figure.”

2. In section 3.1 please explain more about the datasets used such as UCS ultra-conserved elements locus sets.

Response: Thank you for this suggestion. We now have added more details on these datasets.

3. In result section intron, exon, UCE alignment in various real datasets could you please explain the how the percentage gain achieved in CHAPAO, on comparing with four other programs. Have you considered each file bin size for deducing percentage?

Response: We are sorry that it was not clear. The reported performance gains on a particular dataset are for the entire dataset (not for individual bins). We computed the compression gain by considering the entire size of a dataset after it had been compressed using CHAPAO and other methods. We now have made it clear in Section 3. We computed the improvement, in compression gain, of one method over another as follows.

Improvement of method M1 in comparison with another method M2= (size of the file compressed by M2 - size of the file compressed by M1)/size of the file compressed by M2

4. Could you please define how this method will perform if the sequences are highly divergent, and consisting of insertion and deletion which are larger than 50bp.

Response: Since CHAPAO aims for capturing the similarity of the sequences, the performance is expected to degrade for highly divergent sequences. We have already demonstrated the impact of divergence/dissimilarity in our original manuscript (please see Section 3.1 (pages 11,12) and Figure 6). 

As for sequences with larger than 50bp insertions and deletions, we believe that there is no direct association between the quantity of insertion/deletion and the performance of CHAPAO. The length of the sequences, the number of sequences, the divergence between the sequences, and other factors collectively have an impact on CHAPAO's effectiveness. For longer sequences (as in whole genome sequences), 50 bp insertion/deletion should not cause any noticeable problem. However, for shorter sequences, it may affect the performance given the insertions and deletions are in different positions in different sequences of an MSA. But if the indels are aligned (meaning that there is not much divergence in their positions in different MSAs), the performance of CHAPAO should not degrade. Acknowledging the importance of your suggestions, we investigated the amounts of insertions and deletions in the sequences analyzed in our study. On the exon gene set (in the Avian dataset), around 25% of the MSAs contain sequences with (on average) less than 50 bp indels. The average length of these sequences is 831 bp and the compression ratio (compressed size/original size) attained by CHAPAO is 0.0734. The rest of the MSA files in the exon dataset have more than 50 bp indels (per sequence). These sequences are 1876 bp long (on average), and the compression ratio of CHAPAO is 0.059. This may lead to the incorrect conclusion that CHAPAO performs better on MSAs with > 50 bp indels than on MSAs with fewer than 50 bp indels. However, we suspect that this is attributable to the file size and the level of similarity among the sequences, not the number of indels. In comparison to shorter sequences, CHAPAO was able to capture more similarity in longer sequences, resulting in a greater compression ratio (although by a small margin). Therefore, we believe that there is no notable impact of the number of indels on the performance of CHAPAO. However, without more rigorous and systematic analyses and investigation, we do not want to discuss this topic in the manuscript, and so we did not include these results in the revised version.

5. Is this method also considering the protein sequences alignments also?

Response: Thank you for raising this point. Indeed, this is an important thing to mention/discuss. In fact, we have already discussed this in Sec 4 (Conclusions). CHAPAO, in its current form, can handle protein sequences, but CHAPAO is not particularly tailored for protein alphabets. As the number of different characters present in protein sequences is significantly higher than the alphabet size of DNA sequences, special customization is required to handle alignments of protein families. We leave this as future work.

6. Edit the references according to journal format.

Response: We have formatted the references accordingly.

7. Please add author contributions section.

Response: Thank you for this suggestion. We have added an author contributions section.

Minor comments:

English language: The grammar throughout the manuscript needs editing and the language in general needs some streamlining.

Response: We have fixed all the issues that you reported below. Thank you for pointing out these issues. In addition, we have carefully edited the manuscript for grammatical errors. We believe you will find this manuscript more convincing.

Introduction:

1. Page 2, Paragraph-1, Line-2, Reference based should be reference-based.

2. Paragraph-3, Line-4, etc should be etc,

3. Parahraph-4, Line-10, format should be formats. 

4. Paragraph-4, Line-17, last two decades should be the last two decades.

5. Paragraph-6, Line-7, dataset should be datasets.

Methods:

1. Section 2.1, Line-1, reference based should be reference-based.

2. Line 3, substitution should be substitution,

Experimental studies:

1. Paragraph-3, Line-3. Ram should be RAM,

Section 3.1 Results on avian dataset

Paragraph 1, Line-3, set should be set, 

Section 3.4 Impact of sliding window lengths on compression gain.

Paragraph-1, Line-7, tend should be tends.

Paragraph-1, Line-14, additional should be an additional.

Conclusion:

Line 7, reference based should be reference-based.

Line-21, DNA sequence should be DNA sequences.

All supplementary data and figures should be quoted in same format.

Response: We have fixed this.

---

## [Decision Letter · Decision Letter 1]

1 Mar 2022

CHAPAO: likelihood and hierarchical reference-based representation of biomolecular sequences and applications to compressing multiple sequence alignments

PONE-D-21-27907R1

Dear Dr. Bayzid,

We’re pleased to inform you that your manuscript has been judged scientifically suitable for publication and will be formally accepted for publication once it meets all outstanding technical requirements.

Kind regards,

Ram Kumar Sharma, Ph.D

Academic Editor

PLOS ONE

Additional Editor Comments (optional):

Since all the quires has been reasonably responded and necessary suggested changes have been included, current version can be accepted for publication.

Reviewers' comments:

Reviewer's Responses to Questions

**Comments to the Author**

1. If the authors have adequately addressed your comments raised in a previous round of review and you feel that this manuscript is now acceptable for publication, you may indicate that here to bypass the “Comments to the Author” section, enter your conflict of interest statement in the “Confidential to Editor” section, and submit your "Accept" recommendation.

Reviewer #2: All comments have been addressed

2. Is the manuscript technically sound, and do the data support the conclusions?

Reviewer #2: Yes

3. Has the statistical analysis been performed appropriately and rigorously? 

Reviewer #2: Yes

4. Have the authors made all data underlying the findings in their manuscript fully available?

Reviewer #2: Yes

5. Is the manuscript presented in an intelligible fashion and written in standard English?

Reviewer #2: Yes

6. Review Comments to the Author

Reviewer #2: (No Response)

7. PLOS authors have the option to publish the peer review history of their article (what does this mean?). If published, this will include your full peer review and any attached files.

Reviewer #2: No

---

## [Editor Report · Acceptance letter]

29 Mar 2022

PONE-D-21-27907R1 

CHAPAO: likelihood and hierarchical reference-based representation of biomolecular sequences and applications to compressing multiple sequence alignments 

Dear Dr. Bayzid:

I'm pleased to inform you that your manuscript has been deemed suitable for publication in PLOS ONE. Congratulations! Your manuscript is now with our production department. 

Kind regards, 

on behalf of

Dr. Ram Kumar Sharma 

Academic Editor

PLOS ONE